# Characterization of Novel Bacteriophage vB_KpnP_ZX1 and Its Depolymerases with Therapeutic Potential for K57 *Klebsiella pneumoniae* Infection

**DOI:** 10.3390/pharmaceutics14091916

**Published:** 2022-09-10

**Authors:** Ping Li, Wenjie Ma, Jiayin Shen, Xin Zhou

**Affiliations:** 1College of Veterinary Medicine, Institute of Comparative Medicine, Yangzhou University, Yangzhou 225009, China; 2Jiangsu Co-Innovation Center for Prevention and Control of Important Animal Infectious Diseases and Zoonoses, Yangzhou University, Yangzhou 225009, China; 3Joint International Research Laboratory of Agriculture and Agri-Product Safety, The Ministry of Education of China, Yangzhou University, Yangzhou 225009, China; 4National Clinical Research Center for Infectious Diseases, The Third People’s Hospital of Shenzhen, Shenzhen 518112, China

**Keywords:** *Klebsiella pneumoniae*, phage, depolymerase, capsule, biofilm, anti-virulence agent

## Abstract

A novel temperate phage vB_KpnP_ZX1 was isolated from hospital sewage samples using the clinically derived K57-type *Klebsiella pneumoniae* as a host. Phage vB_KpnP_ZX1, encoding three lysogen genes, the repressor, anti-repressor, and integrase, is the fourth phage of the genus *Uetakevirus*, family *Podoviridae*, ever discovered. Phage vB_KpnP_ZX1 did not show ideal bactericidal effect on *K. pneumoniae* 111-2, but TEM showed that the depolymerase Dep_ZX1 encoded on the short tail fiber protein has efficient capsule degradation activity. In vitro antibacterial results show that purified recombinant Dep_ZX1 can significantly prevent the formation of biofilm, degrade the formed biofilm, and improve the sensitivity of the bacteria in the biofilm to the antibiotics kanamycin, gentamicin, and streptomycin. Furthermore, the results of animal experiments show that 50 µg Dep_ZX1 can protect all *K. pneumoniae* 111-2-infected mice from death, whereas the control mice infected with the same dose of *K. pneumoniae* 111-2 all died. The degradation activity of Dep_ZX1 on capsular polysaccharide makes the bacteria weaken their resistance to immune cells, such as complement-mediated serum killing and phagocytosis, which are the key factors for its therapeutic action. In conclusion, Dep_ZX1 is a promising anti-virulence agent for the K57-type *K. pneumoniae* infection or biofilm diseases.

## 1. Introduction

*Klebsiella pneumoniae* (*K. pneumoniae*) is a notorious, Gram-negative, opportunistic pathogen which causes up to 10% of nosocomial infections, such as sepsis, pneumonia, urinary tract infections, surgical site infections and pyogenic liver abscess, especially in people with compromised immune function [1]. The increasing spread of multidrug resistant (MDR) *K. pneumoniae* has limited the selection of antibiotics [2]. In particular, *K. pneumoniae*, carrying the broad-spectrum β-Lactamase (ESBL) [3,4] or carbapenemase [5,6] gene, could cause infections with a high incidence rate and mortality.

*K. pneumoniae* produces many virulence factors that contribute to the pathogenesis, including fimbriae adhesin, siderophores, lipopolysaccharide/O antigen, and capsule polysaccharide/K antigen [7,8]. Compared with classical *K. pneumoniae* (cKpn), hypermucoviscosity *K. pneumoniae* (hmvKpn) has a thicker capsular polysaccharide (CPS) layer, which is specifically manifested in the formation of ≥5 mm viscous fibers when the bacterial colonies are stretched by bacterial loop [9]. It is reported that the high mucus phenotype is critical to the pathogenicity of hypervirulent *K. pneumoniae* (hvKpn) [10,11]. The capsular layer plays a key role in the pathogenicity of *K. pneumoniae*, helping the bacteria against adverse environmental conditions [12], escaping the host immune system [7,13], increasing antibiotic resistance [14,15], and contributing to biofilm formation [16,17]. The highly diverse K capsular of *K. pneumoniae* is encoded by the CPS gene cluster, and its biosynthetic pathway is dependent on Wzx/wzy polymerization [18,19]. So far, at least 79 types of K capsular have been identified, among which K1, K2, K5, K20, K54, and K57 are closely related to invasive infections in humans and animals [20]. In addition to the reports on the K1 and K2 serotype strains that are considered to be the main virulence types [21], the K57 serotype is also considered to be one of the pyogenic liver abscess (PLA)-related strains frequently reported in Asia and the United States [22,23]. The research reports that the K57 serotype *K. pneumoniae* shows similarity to the hypermucoviscous phenotype, with the rmpA carriage and high pathogenicity, but differences in the composition and yield of the capsular polysaccharides to the K1 and K2 serotype [24,25,26].

The phage has attracted the attention of researchers and industry worldwide due to its potential as a substitute for antibiotics because of its ability to infect specific bacteria, but some disadvantages, such as the rapid emergence of anti-phage bacteria, the production of phage antibodies, and the difficulty in formulating clinical application norms, seriously affect the effect of the phage treatment of bacterial infection. Under such circumstance, a variety of enzymes from phages have been reported, such as phage-derived endolysin, virion-associated lysins, and polysaccharide depolymerase [27,28]. Endolysin has a significant bactericidal effect on Gram-positive bacteria that expose the peptidoglycan layer [29,30]. However, the outer membrane of Gram-negative bacteria prevents the penetration and degradation of endolysin [31]. Depolymerase is considered as a potential anti-virulence agent for the prevention and control of Gram-negative biofilm-associated infection because it has been proved that it can effectively degrade capsular polysaccharide (CPS), extracellular polysaccharide (EPS), or lipopolysaccharide (LPS) in the bacterial outer membrane [32,33,34]. Some studies have also reported that depolymerase has significant therapeutic effect in the Galleria mellonella larvae infection model, in the mouse infection model, and in human clinical trials [35,36,37].

At present, various phage depolymerases have been identified for different capsular types of *K. pneumoniae*, such as K1 [38], K2 [39], K3, K21 [13], K5, K8, K30/K69 [40], K13, K22, K37 [41], K11, K21, K25, K30, K35, K64, K69, KN4, KN5 [42], K20, K24 [43], K23 [36], K47 [37], K51 [44], K56, KN1, KN3, KN4 [45], KN2 [46], K57 [47], and K63. Among them, the depolymerases Dep_kpv79 and Dep_kpv767, encoded by the *Klebsiella* phages KpV79 and KpV767, were identified as specific β-galactosidases that cleave the K57-type CPS following the hydrolytic mechanism [47]. More depolymerases that are specific for K57 *K**. pneumoniae* need to be discovered.

In this study, a novel phage vB_KpnP_ZX1 that specifically infected the K57-type of *K. pneumoniae* was isolated. As a new member of the genus *Lastavirus*, the biological and genomic characteristics of vB_KpnP_ZX1 were characterized. A novel depolymerase Dep_ZX1, which is derived from phage vB_KpnP_ZX1, showed a degradation effect on the capsules of *K. pneumoniae*. The activity of Dep_ZX1 in anti-biofilms along with the bactericidal effect, used together with antibiotics, was tested. Based on the promoting effect of Dep_ZX1 on serum killing and phagocytosis, Dep_ZX1 showed a high protective and therapeutic effect on *K. pneumoniae*-infected mice.

## 2. Materials and Methods

### 2.1. Bacteria and Culture Conditions

The *K. pneumoniae* used in this study were isolated from clinical stool samples from Nanjing Cancer hospital (Nanjing, China) in 2021 and cultured in Luria-Bertani (LB) at 37 °C. Wzi and wzc were used for capsule typing [48,49]. *RpoB*, *gapA*, *mdh*, *pgi*, *phoE*, *infB*, and *tonB* were used for Multi-locus sequence typing (MLST) (https://bigsdb.pasteur.fr/klebsiella/, accessed on 1 May 2022). *RmpA*, *aerobactin*, *wabG*, *allS*, *iucB*, *urea*, *fim*, *ybtA*, and *mrKD* were determined for virulence factors. The antibiotic resistance of the bacteria was determined by the standard Kirby–Bauer disk diffusion method of the Clinical Laboratory Standard Institute. All the primers used in this paper are listed in Appendix A.

### 2.2. Phage Isolation and Purification

*K. pneumoniae* 111-2 were used as hosts for screening the potential phages. Phage vB_KpnP_ZX1 was isolated from sewage (Nanjing, China). After three rounds of single plaque purification, the phage particles were enriched by PEG/NaCI, as described previously [50]. The phage suspensions were titrated by a standard soft agar overlay method and stored at 4 °C [51]. The morphology of the phage, through negative staining with 1% phosphotungstic acid, was observed by transmission electron microscope (TEM) JEM-1200EX (jeol Ltd., Tokyo, Japan). The host range specificity of phage vB_KpnP_ZX1 was determined by the standard soft agar overlay method.

### 2.3. Phage Biological Characteristics

The one-step growth curve of phage vB_KpnP_ZX1 was measured to calculate its incubation period and burst size. Phage vB_KpnP_ZX1 was added to 10 mL logarithmic *K. pneumoniae* 111-2 culture (1 × 10^8^ CFU/mL) at a multiplicity of infection (MOI) of 0.1. The mixture was placed at 37 °C for 10 min and centrifuged at 18,500× *g* for 1 min to remove free phages. The precipitate was re-suspended in 10 mL LB broth (time zero) and culture at 37 °C, 220 rpm. Three parallel samples were taken every 10 min and treated with 2% chloroform for 10 min to lyse the cells. The phage titer in the supernatant was determined by the standard soft agar overlay method. Burst size was calculated as the ratio of the count of released phage particles to the count of infected bacterial cells during a single lysis cycle.

The phage (10^8^ PFU/mL) was incubated at 4, 10, 20, 30, 40, 50, 60, 70, and 80 °C for 60 min to test its thermal stability. The phage (10^8^ PFU/mL) was incubated at different pH values (2–11) at 37 °C for 60 min to test its pH stability. The phage titer was determined by the standard soft agar overlay method.

The phage bactericidal curve is very important for its clinical therapeutic value. Phage vB_KpnP_ZX1 was added to logarithmic *K. pneumoniae* 111-2 culture (1×10^8^ CFU/mL) at MOI = 10, 1 and 0.1, cultured at 37 °C and 220 rpm for 8 h. The UV absorption value OD_600_ of the culture was measured by the Smart Microplate Reader (Tecan Austria GmbH, Grodig, Austria) every 30 min. 

### 2.4. Phage Genome Analysis

Phage DNA was extracted by the Virus genomic DNA/RNA Extraction Kit (Tiangen, Beijing, China). The whole genome sequencing of the phage was completed by Sangon Biotech (China, Shanghai). The phage genome sequence was assembled by New Blew 3.0 software.

The putative tRNAs were predicted by tRNAscan-SE 2.0 [52]. The coding domain sequences (CDSs) in the genome were predicted by RAST and alignment by blastX in NCBI [53]. The virulence factors and drug resistance of the genome were searched by Virulence Factors of Pathogenic Bacteria and the Comprehensive Antibiotic Resistance Database [54,55]. The visualization of the phage genome was performed by Easyfig [56]. Blastn and blastp in NCBI were used to the align nucleic acid and amino acid sequences, respectively. Dot plots of whole-genome analysis were generated using Gepard [57]. The phylogenetic analysis of the phage terminase large subunit (ORF48) and the tail fiber protein (TFP)/depolymerase Dep_ZX1 (ORF60) was constructed by MERGA5.0 by the Neighbor Joining method with 1000 bootstraps. The comparative genomic analysis of the phage was carried out by Easyfig. The prediction of the conservative domain and three-dimensional structure was carried out through the Phyre2 server [58].

### 2.5. Depolymerase Dep_ZX1 Expression and Purification

The DNA of depolymerase Dep_ZX1 was ligated to vector pET-28a by homologous recombination. The primers TFP-F/TFP-R and 28A-F/28A-R were used for PCR amplification to obtain linear DNA fragments. The positive plasmid pET28a-TFP was obtained by combining the two DNA fragments with the pEASY-Uni seamless cloning and assembly kit (TransGen, Beijing, China). After being identified by Sanger sequencing, the plasmid pET28a-TFPs were extracted by the Mini prepare plasma kit (Vazyme, Nanjing, China) and transformed into competent cell *E. coli* BL21 (DE3). Isopropyl-β-D-thiogalactopyranoside (IPTG) with a final concentration of 1 mM was added in order to express recombinant depolymerase. After being cultured at 16 °C for 24 h, the bacteria were enriched and resuspended in 20 mL binding buffer (50 mM NaH2PO4; 300 mM NaCl; 10 mM imidazole; pH 7.4). After being lysed by sonication, the supernatant was filtered by 0.22 μm filter and stored at 4 °C. 

Recombinant depolymerase Dep_ZX1, carring a C-terminal-his tag, was purified using the Ni-NTA 6FF gravity column (Sangon, Shanghai, China). Briefly, 20 mL of sterile water was added to clean the column; 20 mL binding buffer was added to balance the gravity column; 20 mL of bacterial lysate supernatant was added to combine with the gravity column; 20 mL of binding buffer with 80 mM imidazole was added to wash the gravity column; and 10 mL of binding buffer with 300 mM imidazole was added to elute the recombinant protein. The eluent was dialyzed overnight to remove free imidazole and concentrated through a 10 kDa ultrafiltration tube (Millipor, Burlington, MA, USA). The purified Dep_ZX1 was quantified by the BCA protein concentration assay kit (Beyotime, Nantong, China) and stored at −80 °C. The protein samples were analyzed by Sodium Dodecyl Sulfate Polyacrylamide Gel Electrophoresis (SDS-PAGE) with Coomassie Brilliant Blue (Beyotime, Nantong, China).

### 2.6. Anti-capsule Activity of Depolymerase

Depolymerase Dep_ZX1 was diluted to 100, 10, 1, 0.1, 0.01, 0.001, and 0.0001 μg/mL with PBS. The activity of the Dep_ZX1 was determined by spot test. Briefly, 2 μL of Dep_ZX1 (100^−0.0001^ μg/mL) or PBS was spotted onto freshly *K. pneumoniae* 111-2 lawns and incubated for 6–8 h at 37 °C. 

The *K. pneumoniae* 111-2 colonies cultured overnight on the LB plate were suspended in PBS buffer. Then, the Dep_ZX1 with a final concentration of 10 μg/mL was added and incubated at 37 °C for 2 h, with PBS as a blank control. The bacteria were washed twice with PBS and fixed with 2.5% glutaraldehyde for 3 h. The bacteria were stained with 2% phosphotungstic acid and observed by transmission electron microscope. 

### 2.7. Phage Adsorption

The competitive binding of depolymerase Dep_ZX1 and phage was as follows: Logarithmic *K. pneumoniae* 111-2 (10^8^ PFU/mL) was resuspended in PBS buffer. Phage vB_KpnP_ZX1 with a final titer of 10^7^ PFU/mL was added to all groups. Then, the Dep_ZX1 with a final concentration 10 μg/mL was added to a competitive group, with PBS as a negative control. After incubation at 37 °C for 10 min, the phage titers in the supernatant were determined by the standard soft agar overlay method.

The inhibition of the phage adsorption by depolymerase Dep_ZX1 was as follows: Logarithmic *K. pneumoniae* 111-2 (10^8^ PFU/mL) was treated with Dep_ZX1 (10 μg/mL) at 37 °C for 1 h, with PBS as a negative control. Then, the host bacteria were washed by PBS three times. Phage vB_KpnP_ZX1 with a final titer of 10^7^ PFU/mL was added to all groups. After incubation at 37 °C for 10 min, the phage titers in the supernatant were determined by the standard soft agar overlay method. 

### 2.8. Stability of Depolymerase

The depolymerase Dep_ZX1 (100 µg/mL) was incubated at 4, 30, 40, 50, 60, and 70 °C for 60 min and then cooled down on ice for 2 min to evaluate the thermal stability. Similarly, the pH stability of Dep_ZX1 was evaluated after incubation for 60 min in different pH buffers (4–11). The Dep_ZX1 treated as above was diluted to 100, 10, 1, and 0.1 µg/mL with PBS. Then, 2 μL Dep_ZX1 was spotted onto freshly *K. pneumoniae* 111-2 lawns and incubated for 6–8 h at 37 °C 

### 2.9. Antibiofilm Activity of Depolymerase

Depolymerase Dep_ZX1 inhibits the formation of biofilm: Logarithmic *K. pneumoniae* 111-2 was resuspended in 2 × TSB. The Dep_ZX1 was diluted to 200, 20, 2, 0.2, and 0.02 μg/mL with PBS. Then, 100 μL *K. pneumoniae* 111-2 and 100 μL Dep_ZX1 with a final concentration of 100^−0.01^ μg/mL were added to different wells in 96-well flat-bottomed polystyrene microtiter plates (Corning, New York, NY, USA), with PBS as a negative control. After being cultured at 37 °C for 48 h without shaking, each well was washed by PBS twice. Crystal violet was used to determine the content of the biofilm. Briefly, the biofilm was fixed by methanol for 20 min, stained by 1% crystal violet for 10 min, and dissolved by 33% acetic acid for 10 min. The optical density OD_595_ was measured by the Smart Microplate Reader (Tecan Austria GmbH, Grodig, Austria). 

Depolymerase Dep_ZX1 disrupts the formed biofilm: The biofilm in the 96-well plate, formed by 200 μL of logarithmic *K. pneumoniae* 111-2, was cultured for 48 h at 37 °C. After being washed twice by PBS, 100 μL of Dep_ZX1 with a final concentration of 100^−0.01^ μg/mL was added for incubation at 37 °C for 2 h, with PBS as a negative control. Each well was washed with PBS, and the content of the biofilm was determined with crystal violet staining. 

### 2.10. Antibacterial Activity of Depolymerase

The minimum inhibitory concentration (MIC) of kanamycin, gentamicin, and streptomycin on the planktonic cells of *K. pneumoniae* 111-2 was determined. It was found that the MICs of kanamycin, gentamicin, and streptomycin were all 0.5 μg/mL. The mature biofilm was formed by 200 μL of logarithmic *K. pneumoniae* 111-2, growing in 96-well plates for 48 h, and then washed by PBS 3 times. When the mature biofilms were treated with antibiotics (0.5, 1, 2, 4 μg/mL) for 2 h, a significant decrease in bacterial count was observed at the 4 μg concentrations of kanamycin/gentamicin/streptomycin. 

In order to obtain a suitable combination of Dep_ZX1(10 μg/mL) and antibiotics (2, 4, 8, and 16 μg/mL) that kills bacteria with biofilms, the mature biofilm was treated at 37 °C for 2 h in the combined system. The biofilm was divided into four groups: (1) the control group, treated with PBS for 2 h; (2) the depolymerase group, treated with Dep_ZX1(10 μg/mL) for 2 h; (3) the antibiotic group, treated with antibiotics for 2 h; and (4) the combined group, treated with antibiotics and Dep_ZX1 together for 2 h. After treatment, the biofilm was cleaned, scraped off, and dissolved in 200 μL PBS. The mixture was diluted gradiently and then, the number of living cells on the culture dish was estimated. 

### 2.11. Phagocytosis Assay

Mouse peritoneal macrophage RAW264.7 was used as the cell model to test the phagocytic effect of immune cells on *K. pneumoniae* 111-2. Logarithmic *K. pneumoniae* 111-2 (5 × 10^8^ CFU/mL) were resuspended in DMEM medium after treatment with Dep_ZX1 (10 μg/mL) at 37 °C for 2 h, with PBS as a negative control. The cells were cultured in DMEM medium containing 10% fetal bovine serum to form a single cell layer in a 24-well plate. The cells were washed with PBS three times, and 500 μL of treated *K. pneumoniae* 111-2 was added according to the ratio of bacteria cells of 10:1 and incubated at 37 °C for 2 h. After washing, the cells were incubated with DMEM medium containing kanamycin (100 μg/mL) at 37 °C for 2 h. Finally, the cells were washed again and resuspended in 1 mL sterile water (repeat blow to ensure the cells were completely lysed). The number of bacteria was measured, and the anti-phagocytosis was calculated. 

### 2.12. Serum Killing Assay

Serum was taken from SPF female mice aged 6 weeks and filtered through a 0.22 filter. The logarithmic *K. pneumoniae* 111-2 (10^6^ CFU/mL) were resuspended in PBS after treatment with Dep_ZX1 (10 μg/mL) at 37 °C for 2 h, with PBS as a negative control. After the bacteria were washed with PBS three times, the same volume of serum, complement inactivated serum (56 °C, 30 min), or PBS was added and cultured at 37 °C for 2 h. The number of bacteria was immediately determined.

### 2.13. Therapeutic Test of Depolymerase in Mice

Specific-pathogen-free female Male BABL/C mice (18–20 g) aged 6 weeks were purchased from the Experimental Animal Center of Yangzhou University. All the animal experiments were performed in strict accordance with the Animal Welfare and Research Ethics Committee of Yangzhou University. The minimal lethal dose (MLD) of *K. pneumoniae* 111-2 in mice is 10^8^ CFU.

The mice were divided into four groups (10 mice in each group). Control group N: the mice were intraperitoneally injected with PBS (200 µL). Infection group: the mice were intraperitoneally infected with MLD of *K. pneumoniae* 111-2 (100 µL), then PBS (100 µL) was injected 1 h later. Prevention group: the mice were intraperitoneally injected with 50 μg Dep_ZX1 (100 µL), then *K. pneumoniae* 111-2 (100 µL) were injected 1 h later. Treatment group: the mice were intraperitoneally infected with 10^8^ CFU of *K. pneumoniae* 111-2 (100 µL), then 50 µg Dep_ZX1 (100 µL) was injected 1 h later. The survival number of the mice was monitored for 7 days. The blood, lungs, livers, spleens, and kidneys of the mice were collected 12/24 h after bacterial infection for bacterial count. The lungs, livers, spleens, and kidneys of the mice were collected 48 h after bacterial infection for histological analysis. The organs were fixed with 4% paraformaldehyde solution, dehydrated with ethanol, embedded in paraffin, sliced, and stained with hematoxylin and eosin (H&E). Further observation and histological analysis were performed under light microscope.

### 2.14. Statistical Analysis

The experiments of the phage biological characteristics, phage adsorption assay, biofilm assay, bacterial survival assay, macrophage phagocytosis assay, serum kill assay, depolymerases activity, and stability assay were conducted in triplicate. All statistical analyses in this study were carried out using GraphPad Prism 5 software. An unpaired *t* test was used for significance analysis.

## 3. Results

### 3.1. Phage vB_KpnP_ZX1

The *K. pneumoniae* 111-2 were identified as Wzi206, the K57 capsule type, and the ST412 MLST type. *RmpA*, *aerobactin*, *wabG*, *allS*, *iucB*, *urea*, *fim*, *ybtA*, *mrKD* were detected in the *K. pneumoniae* 111-2. *K. pneumoniae* 111-2 has the hypermucoviscous phenotype (string ≥ 5 mm) on the blood agar plate (Appendix A). The antibiotic susceptibility test showed that *K. pneumoniae* 111-2 was resistant to ampicillin, sulfamethoxazole, tobramycin, azithromycin, and erythromycin (Appendix A). 

Phage vB_KpnP_ZX1 forms a turbid round plaque with a halo around it on the bacterial lawn of *K. pneumoniae* 111-2 (Figure 1a). TEM showed that phage vB_KpnP_ZX1 possess an isometric head of 55 ± 0.5 nm in diameter and a short tail (Figure 1b). The host range analysis of the phage showed that phage vB_KpnP_ZX1 could only infect the K57-type *K. pneumoniae* 111-2 (Table 1).

### 3.2. Biological Characteristics of Phage vB_KpnP_ZX1

The one-step growth curve of the phage revealed that vB_KpnP_ZX1 had a latent period of approximately 30 min and a burst size of about 125 PFU/cell (Figure 2a).

The thermal and pH stability analysis shows that vB_KpnP_ZX1 maintained a medium tolerance range. The phage vB_KpnP_ZX1 viability shows a high stability after being treated at 4–40 °C for 1 h. A decrease appeared at 50 °C and disappeared at 60 °C (Figure 2b). The phage viability remained stable after being treated in a pH buffer (5–10) for 1 h. A decrease was observed at pH = 3/4/10/11 and disappeared at pH = 2/11 (Figure 2c). 

The phage bactericidal curve shows that vB_KpnP_ZX1 has a slow bactericidal speed, and the number of bacteria shows an increase–decrease–increase trend (Figure 2d). For example, when the phages were added at the ratio of MOI = 10, the growth trend of the bacteria was inhibited in the first 60 min; the number of bacteria sharply declined from 60–120 min and increased again after 120 min. Although phage vB_KpnP_ZX1 encodes an integrase, we did not isolate lysogenic *K. pneumoniae* 111-2 during the propagation of the phage. Therefore, we speculate that the increase in the number of bacteria at the later stage of phage vB_KpnP_ZX1 infection is caused by the production of phage-resistant strains rather than the lysogenesis of the bacteria.

### 3.3. Genome Analysis of vB_KpnP_ZX1

The genome of phage vB_KpnP_ZX1 is a dsDNA that comprises 60,982 bp with a GC content of 57%. One tRNA was predicted. No virulence gene or drug resistance gene was detected. Gp23 repressor, gp28 anti-repressor, and gp77 integrase were annotated as lysogen-related genes. A total of 78 open reading frames (ORFs) were annotated in phage vB_KpnP_ZX1, of which 30 ORFs were functional proteins and 48 ORFs were hypothetical proteins (Appendix A). The main ORFs of the phage are associated with nucleic acid metabolism (ORF9, ORF17, ORF19, ORF23, ORF24, OR25, ORF26, ORF29, ORF37, ORF41, ORF42, ORF43, ORF47, OR52, ORF59, ORF67, ORF71, and ORF77); phage structural (ORF50, ORF53, ORF60, ORF64, and ORF68); phage assembly (ORF46 and ORF48); and phage lysis (ORF72, ORF73, and ORF75) (Figure 3a). Most functional genes were transcribed in the same direction as the same DNA strand. 

Phage vB_KpnP_ZX1 only has sequence similarity with MF612073.1 *Klebsiella* phage SopranoGao (73%), MT251348.1 *Klebsiella* phage SJM3 (64%), and NC_054965.1 *Klebsiella* phage LASTA (64%) and less than 3% with the other phages (Figure 3b). The whole-genome dot plotting analysis shows that 16 phages from different genera were classified into 5 clusters, within which the phages vB_KpnP_ZX1, SopranoGao, SJM3, and LASTA belong to the *Podoviridae* family, *Lastavirus* genus (Figure 3c). In addition, the phylogenetic analysis of the phage terminase large subunit confirms the same result (Figure 3d). 

Phage vB_KpnP_ZX1 has a similar genome size (62 kb), GC content (49%), and gene structure to SopranoGao, SJM3, and LASTA. These four phages encode a large molecular protein PLxRFG (495 KDa) that may be involved in DNA metabolism, although the PLxRFG protein of phage vB_KpnP_ZX1 is significantly different from the other three phages. Figure 3b shows that the biggest difference between the phages vB_KpnP_ZX1 and SopranoGao, SJM3, and LASTA was the DNA metabolism-related proteins and TFP. Although lysogen-related genes have been annotated in the phages vB_KpnP_ZX1, SopranoGao, SJM3, and LASTA, they are not completely consistent. The integrase of phage vB_KpnP_ZX1 (ORF 77) has 95% similarity with that of phage SopranoGao and only 20% similarity with that of SJM3 and LASTA. No repressor or anti-repressor was annotated in the phages SopranoGao, SJM3, and LASTA. However, the transcription regulator (ORF53) of the phage SopranoGao has 98% similarity with the repressor (ORF23) of the phage vB_KpnP_ZX1. TFP is extremely important in the early recognition and adsorption of phage [59,60,61]. The low similarity of TFP suggests differences in the adsorption receptor and host range between phage vB_KpnP_ZX1 and the others. Phage vB_KpnP_ZX1 can specifically infect K57-type *K. pneumoniae* in this study. Unfortunately, the host range of the phages SopranoGao, SJM3, and LASTA has not been reported so far.

### 3.4. Sequence Analysis of Depolymerase 

Blasp aliment showed that the TFP of phage vB_KpnP_ZX1 had the highest homology with the TFP of *Klebsiella* phage vB_KpnM_KpV79 (coverage 94%, identity 89.35%), followed by the TFP of *Enterobacter* phage ENC14 (coverage 91%, identity 55.93%), and *Klebsiella* phage vB_KpnP_KpV767 (coverage 91%, identity 55.93%). The phylogenetic tree showed that the TFP of phage vB_KpnP_ZX1 and vB_KpnM_Kpv79 belong to the same branch (Figure 4a). TFP/Dep_kpv79, encoded by phage vB_KpnM_Kpv79, and TFP/Dep_kpv767, encoded by phage vB_KpnP_Kpv767, have been identified as β-Galactosidase, which can specifically degrade the K57 capsular polysaccharide of *K. pneumoniae* [47]. The structural bioinformatics analysis shows that depolymerase include three functional domains: the N-terminal domain is responsible for connecting receptor binding proteins to the phage tail (anchor domain); the central helical domain has proteinase activity; and the C-terminal auto-cleavable chaperone domain participates in host cell recognition [62,63,64]. The N-terminal amino acid of Dep_ZX1 and Dep_kpv79 has low consistency, while the central domain and C-terminal amino acid sequences have high consistency (Figure 4b). The difference of species may be the reason for the great difference in the N-terminal domain; that is, phage vB_KpnM_Kpv79 belongs to the *Myoviridae* family, *Jedunavirus* genus, and phage vB_KpnP_ZX1 belongs to the *Podoviridae* family, *Lastavirus* genus. The high similarity between the central domain and the C-terminal may indicate that Dep_ZX1 may degrade the K57 capsular of *K. pneumoniae* with β Galactosidase activity as well. 

Phyre2 shows that 6% of the alpha helix (grey) and 49% of the beta strand (blue) were predicted in the secondary structure of Dep_ZX1 (Figure 4c). The 61–459 amino acids of Dep_ZX1 have homology with α-1,3-glucanase (confidence, 99.5%; coverage, 64%); the pectin lyase-like superfamily protein (confidence, 99.5%; coverage, 56%); and exo-poly-α-d-galacturonidase (confidence, 99.6%; coverage, 57%). Here, the simulated 3D structure of 64–459 amino acids in Dep_ZX1 was based on the sequence of α-1,3-glucanase (confidence, 99.5%; identity, 22%) (Figure 4d). Studies have shown that the slender and highly intertwined β-helical domains form specific catalytic pockets in depolymerase [63]. The pocket structure was predicted at 80–290 amino acids in Dep_ZX1 (Figure 4e). Therefore, 64–459 amino acids in Dep_ZX1 may be responsible for capsule degradation activity. The possible result is that 64-459 amino acids in Dep_ZX1 form a three-sided parallel helical structure responsible for polysaccharide degradation, and 80–290 amino acids form catalytic pockets. 

### 3.5. Capsule Degradation Activity of Depolymerase

Depolymerase Dep_ZX1 was cloned into plasmid pET-28a, and the agarose gel electrophoresis results of the PCR products using primer T7-F/T7-R are shown in Figure 5a. The purified Dep_ZX1 migrated as a single band with a molecular weight of about 70 kDa on SDS-PAGE gel (Figure 5b). The Western blot image of the purified depolymerase Dep_ZX1 is show in Appendix A.

The degradation activity of Dep_ZX1 against *K. pneumoniae* 111-2 was determined by drop test. The results showed that Dep_ZX1 produced a degradation halo on the bacterial lawn of *K. pneumoniae* 111-2, but the PBS buffer did not. The lowest concentration of Dep_ZX1 with significant activity is 0.1 μg/mL (Figure 5c). TEM showed that the rough surface of logarithmic *K. pneumoniae* 111-2 is surrounded by minute pili of about 1 μm and a thick capsule of about 100 nm (Figure 5d). After Dep_ZX1 treatment, the “capsule-stripped” bacteria were generated (Figure 5e). Compared with the logarithmic phase, the surface of *K. pneumoniae* 111-2 in the mature phase has a thicker capsule of about 200 nm and less pili (Figure 5f). This may be caused by the thickening of capsule interfering with the growth of the pili [65]. After Dep_ZX1 treatment, the surface of the bacteria becomes smooth too (Figure 5g). These results indicate that Dep_ZX1 has a high degradation effect on the capsule of *K. pneumoniae* 111-2.

### 3.6. Stability of Depolymerase

The temperature stability test showed that the minimum effective concentration of Dep_ZX1 in treatment at 40, 50, and 60 °C for 1 h, forming a degradation halo on *K. pneumoniae* 111-2, was 0.1, 0.1, 100 µg/mL, respectively (Table 2). The pH stability test showed that the minimum effective concentration of Dep_ZX1 in treatment at pH 4–9 for 1 h was 0.1 µg/mL and of Dep_ZX1 in treatment at pH 3/10 for 1 h was 10 µg/mL (Table 2). The conditions for keeping high activity of Dep_ZX1 are temperature < 50 °C and pH = 4–9.

### 3.7. Depolymerase Inhibits Phage Adsorption

The competitive binding test showed that more than 99.5% of phage vB_KpnP_ZX1 were adsorbed in the surface of *K. pneumoniae* 111-2 within 10 min. However, only 66.6% of the phages were adsorbed by *K. pneumoniae* 111-2 when adding with Dep_ZX1 simultaneously. This result indicates that Dep_ZX1 can specifically bind to the surface of *K. pneumoniae* 111-2 (Figure 6a). In addition, only 53.3% of the phages were adsorbed by *K. pneumoniae* 111-2 after treatment with Dep_ZX1 (Figure 6b). The treatment of Dep_ZX1 reduces the cell capsule; so, the capsules play an important role in the adsorption of phage vB_KpnP_ZX1.

### 3.8. Antibiofilm Activity and Antibacterial Activity of Depolymerase

Figure 7a,b shows that Dep_ZX1 with a concentration of 0.1^−100^ μg/mL can significantly reduce the formation of biofilm and degrade the formed biofilm. Dep_ZX1 cannot kill *K. pneumoniae* 111-2, although it can degrade its capsule. When used in combination with antibiotics, Dep_ZX1 increases bacterial susceptibility to antibiotics (Figure 7c). Gentamicin, streptomycin, or kanamycin with a concentration 4 μg/mL can significantly reduce the bacteria count with mature biofilms. When combined with Dep_ZX1 (10 μg/mL), gentamicin (2 μg/mL), streptomycin (2 μg/mL), or kanamycin (2 μg/mL) can significantly reduce the bacteria count with mature biofilms. The minimum bactericidal concentrations (MBCs) of gentamicin, streptomycin, and kanamycin against bacteria with mature biofilms were 8, 16 and 32 μg/mL, respectively. When antibiotics and Dep_ZX1 act together, lower concentration of antibiotics (gentamicin, 4 μg/mL; streptomycin, 8 μg/mL; kanamycin, 8 μg/mL) can kill 99.9% of bacteria. The bacterial concentration with mature biofilms reduces to 10^3^ CFU/mL after treatment by gentamicin, streptomycin, and kanamycin (16 μg/mL) combined with Dep_ZX1 (10 μg/mL) for 2 h.

### 3.9. Depolymerase Improves K. pneumoniae Sensitivity to Complement-Mediated Killing and Phagocytosis

Depolymerase’s capsule degradation effect on *K. pneumoniae* 111-2 significantly increased the phagocytosis of the mouse peritoneal macrophage and the bactericidal efficiency of the mouse serum. Compared with *K. pneumoniae* 111-2, the phagocytosis efficiency of the macrophages on *K. pneumoniae* 111-2 treated with Dep_ZX1 increased from 23% to 63% (Figure 8a). The serum can kill 50% of the *K. pneumoniae* 111-2, but the complement-inactivated serum loses its bactericidal effect (Figure 8b). The bactericidal effect of mouse serum on *K. pneumoniae* 111-2 treated with Dep_ZX1 reached 95%. Similarly, the complement-inactivated serum also lost its bactericidal effect on *K. pneumoniae* 111-2 after Dep_ZX1 treatment. This means that Dep_ZX1 can increase the bactericidal effect of the complement-mediated serum.

### 3.10. Depolymerase Improves the Survival Rate of K. pneumoniae-Infected mice

All the mice in the challenge group were dead within 24 h after intraperitoneal injection of *K. pneumoniae* 111-2 at the dose of 10^8^ CFU. The number of bacteria in the blood of the mice at 12 h and 24 h after infection was 10^6^ and 10^8^ CFU/mL, respectively. Significantly, no mice died in the prevention group, injecting intraperitoneally with 50 μg of Dep_ZX1 before or after infection with *K. pneumoniae* 111-2 (Figure 9a). The number of bacteria in the blood of the mice in the prevention group or the treatment group was 10^3^ and 0 CFU/mL after 12 h and 24 h of infection (Figure 9b). H&E staining of the tissue showed the infiltration of inflammatory cells in the lungs, kidneys, and livers, expansion of the alveolar ducts, expansion of the hepatic sinuses, and necrosis of the small intestinal villi in the infected group. The pathological changes in the mouse organs in the prevention group and the treatment group were significantly reduced (Figure 9c). The results showed that the Dep_ZX1 had an effective preventive and therapeutic effect on the mice infected by *K. pneumoniae* 111-2. The degradation of bacterial polysaccharides by Dep_ZX1 leads to the increase in phagocytosis and serum killing, which is the main reason for the rapid elimination of bacteria by the immune system.

## 4. Discussion

The first *Lastavirus* phage, SopranoGao, was isolated from sewage samples in the United States (North America) in 2017; then, the second and third *Lastavirus* phages, SJM3 and LASTA, were isolated from sewage samples in Serbia (Europe) in 2020. In this article, a novel phage, vB_KpnP_ZX1, of the *Lastavirus* genus was isolated from sewage samples in China (Asia) in 2021. The genomic characteristics of the *Lastavirus* genus were analyzed for the first time. Similar genome size, GC content and, gene structure exist in the genome of the phages vB_KpnP_ZX1, SopranoGao, SJM3, and LASTA. All these four phages encode a large molecular protein PLxRFG (495 KDa) and a lysogenic-related gene. As the phages encode a gene accounting for such a large proportion (19%), we are curious to know whether it is detrimental to the phage proliferation efficiency. The biological characteristics of the phage vB_KpnP_ZX1 show that it has a long incubation period of 30 min, a medium burst size of 125 PFU/cell, a wide pH stability of 5–10, and a low temperature tolerance of 0–40 °C. The long incubation period is unfavorable to the bactericidal, which may be the reason why the bactericidal effect of bacteriophage vB_KpnP_ZX1 is not ideal. The rapid production of phage-resistant strains and the existence of lysogen-related genes indicate that the phage vB_KpnP_ZX1 is not suitable for clinical treatment.

Depolymerases are generally considered to be structural proteins, located in/at the tail fiber, substrate, and neck of the phages. Most phages encode one depolymerase, but a few phages can encode a variety of depolymerases. For example, the *Escherichia coli* phage K1-5 encodes two depolymerases that degrade the K1 and K5 polysaccharide capsules [66]; *Bacillus subtilis* Myophage CampHawk encodes a peptidase and three pectate lyase domains [67]; and *Klebsiella* Phage ΦK64-1 encodes 11 kinds of depolymerases for multiple host capsular types [42]. According to reports, depolymerases usually exist in the form of trimers and have a patterned structure [68]. The conserved N-terminal anchor domain that facilitates phage self-assembly, a K-type variable center β-helical domain for polysaccharide recognition and catalysis, is composed of a C-terminal domain responsible for folding, trimerization, and receptor recognition [27,44,63]. According to their mode of action, the phage depolymerases were divided into two main classes: hydrolases and lyases [69]. Hydrolases degrade O-antigen side chains of peptidoglycans, capsular polysaccharides, or LPS by catalyzing the cleavage of the glycosyl oxygen bonds in the glycosidic bonds [70]; these include sialidase [71], rhamnosidase [72], levosanase [73], glucanase [74], xylan [75], and LPS deacetylase [76]. Lyase uses the β-elimination to introduce double bonds between C4 and C5 of non-reducing uronic acid after the glycosidic bond between monosaccharide and C4 of uronic acid breaks [77]; these include alginate lyase [78], hyaluronate lyase [79], pectinate lyase [80], K5 lyase [66], and O-specific polysaccharide lyase [81]. The capsular polysaccharide of *Klebsiella* K57 serotype is composed of branched tetrasaccharide repeats, which contain two D-mannose residues, as well as one D-galactose and D-galacturonic acid residues [82,83]. Depolymerases Dep_kpv79 and Dep_kpv767 are β-glycosidases that cleave specifically the β-d-Galp-(1→3)-d-GalpA (A→C) linkages in the CPS of *K. pneumoniae* KPi8289 by the hydrolytic mechanism [47]. Based on the high similarity between the central and C-terminal domains of Dep_ZX1 and Dep_kpv79, we speculate that Dep_ZX1 may also be β-Galactosidase. Although this is not the first discovery of the K57-type depolymerase, it may be helpful to study the catalytic domain of these depolymerases in future research. 

Biofilm refers to the complex, sessile communities of microbes found either attached to a surface or buried firmly in an extracellular matrix as aggregates [84]. The biofilm matrix surrounding bacteria makes them tolerant to harsh conditions and resistant to antibacterial treatments and also causes them to disperse and colonize new niches [84,85]. Biofilm disease includes device-related infections, chronic infections in the absence of a foreign body, and malfunction of medical devices, which are difficult to effectively treat [86]. Certain kinds of depolymerases, including Dep_ZX1, can effectively prevent the formation of biofilm, and the degradation of biofilm has been formed. In fact, depolymerase can degrade the surface decorative polysaccharides of bacteria instead of killing bacteria. According to reports, the reduction in polysaccharides is conducive to the penetration of antibiotics, particularly positively charged aminoglycosides through biofilm [87]. For example, depolymerase can enhance the sensitivity of polymyxin, gentamicin, and ciprofloxacin to *K. pneumoniae* in biofilm [88,89,90]. In this paper, Dep_ZX1 can enhance the sensitivity of gentamicin, streptomycin, and kanamycin to *K. pneumoniae* in biofilm. The combination of Dep_ZX1 (10 μg/mL) and gentamicin (4 μg/mL), streptomycin (8 μg/mL), and kanamycin (8 μg/mL) can kill 99.9% of bacteria. Phage depolymerase has shown good effects in inhibiting a variety of biofilms, which makes it possible for it to become an effective weapon against catheter-related infection [91]. 

Based on the degradation of capsular polysaccharide by depolymerase, it has shown a therapeutic effect in the treatment of infection caused by *K. pneumoniae* with a high mucus phenotype. It has been reported that mice and Galleria mellonella larvae infected with *K. pneumoniae* were successfully rescued by depolymerase [36,37,92,93]. In this paper, Dep_ZX1 had an effective preventive and therapeutic effect on bacteremia caused by *K. pneumoniae* 111-2 infection. In the prevention and treatment experiments with 50 µg Dep_ZX1, the mortality of the mice was reduced from 100% to 0. The therapeutic principle of depolymerases is that they weaken the bacterial defense against the immune system by cleaving the capsular polysaccharide of *K. pneumoniae*, such as by complement-mediated killing and the phagocytosis of macrophages [13,47,93]. In this paper, Dep_ZX1 also has the effect of improving the phagocytic capacity of phagocytes and the bactericidal capacity of the serum. 

## 5. Conclusions

A novel phage vB_KpnP_ZX1 of the *Lastavirus* genus was isolated in this paper. The biological and genomic characteristics and depolymerase Dep_ZX1 of phage vB_KpnP_ZX1 were characterized. Our results showed that Dep_ZX1 had high degradation activity on the K57-type *K. pneumoniae* capsule. As the protective structure of bacteria, the capsule can increase the virulence of bacteria and hinder the penetration of antibiotics. From an antibacterial test, we confirmed that the combination of Dep_ZX1 and antibiotics (gentamicin, streptomycin, and kanamycin) can improve the antibiotic sensitivity of *K. pneumoniae*. Furthermore, a single dose of Dep_ZX1 can degrade mature biofilm and save *K. pneumoni**ae*-infected mice. All the data show that the combination of Dep_ZX1 and antibiotics is expected to become a candidate drug in the clinic for controlling *K. pneumoniae* infection or catheter-related infection caused by biofilm.

## Figures and Tables

**Figure 1 pharmaceutics-14-01916-f001:**
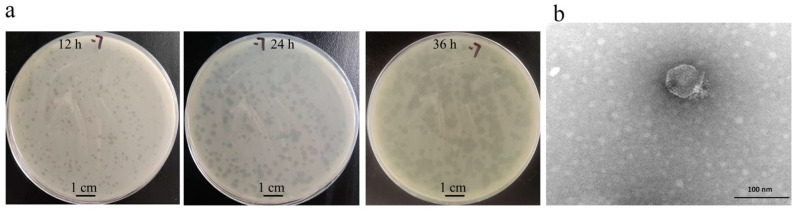
Phage morphology: (**a**) plaque and (**b**) transmission electron microscope of phage vB_KpnP_ZX1.

**Figure 2 pharmaceutics-14-01916-f002:**
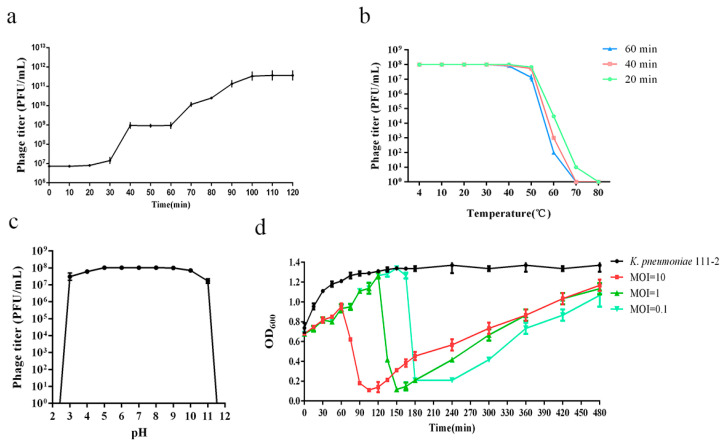
Biological characteristics of phage vB_KpnP_ZX1: (**a**) one-step growth curve; (**b**) thermal stability; (**c**) pH stability; (**d**) phage bactericidal curve.

**Figure 3 pharmaceutics-14-01916-f003:**
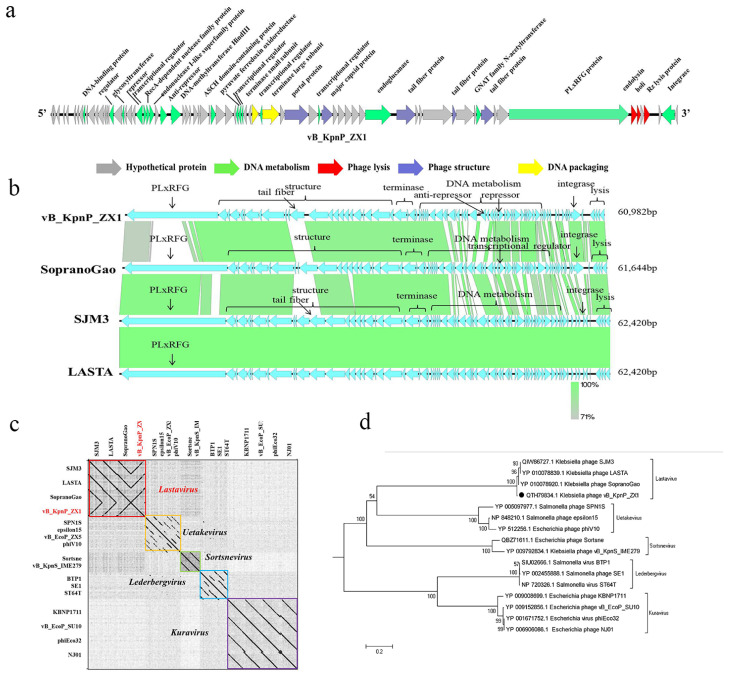
Genomic analysis of phage vB_KpnP_ZX1. (**a**) The genomic characteristics of vB_KpnP_ZX1. (**b**) Comparative genomic analysis of phages vB_KpnP_ZX1, SopranoGao, SJM3, and LASTA. (**c**) Genome dot plot analysis of the phage vB_KpnP_ZX1. Different genome clusters are marked with different colored boxes. (**d**) Phylogenetic tree of the terminase large subunit of the phage vB_KpnP_ZX1.

**Figure 4 pharmaceutics-14-01916-f004:**
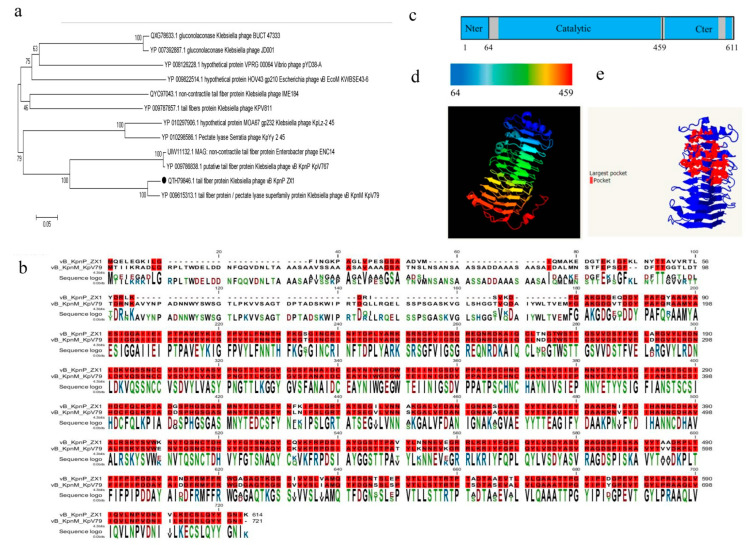
The amino acid sequences analysis of depolymerase Dep_ZX1. (**a**) Phylogenetic tree of Dep_ZX1. (**b**) Amino acid alignment between Dep_ZX1 and Dep_kpv79. The same amino acids are represented by a red background. (**c**) The secondary structure diagram of Dep_ZX1; gray represents alpha helix, and blue represents beta strand. (**d**) Simulated three-dimensional structure of intermediate domain in Dep_ZX1. (**e**) Pocket structure of Dep_ZX1.

**Figure 5 pharmaceutics-14-01916-f005:**
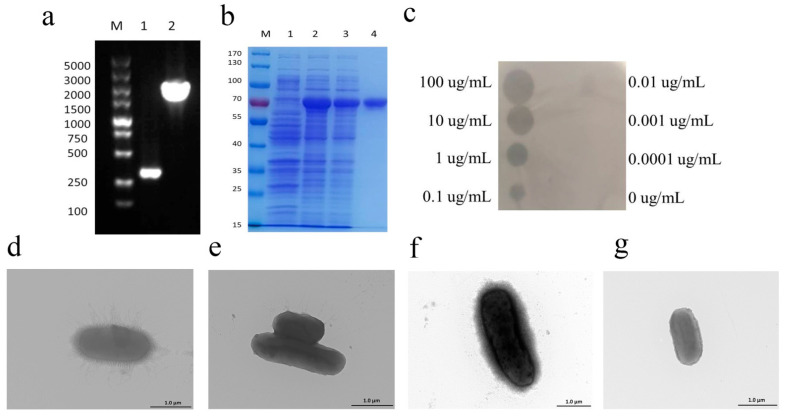
The expression and activity test of recombinant depolymerase Dep_ZX1. (**a**) Agarose nucleic acid electrophoresis picture of Dep_ZX1 expression plasmid; 1: negative control, pET-28a; 2: positive plasmid, pET-28a-TFP. (**b**) The SDS-PAGE image of purified depolymerase Dep_ZX1; 1: *E. coli* BL21(DE3) carrying pET-28a plasmid expressed for 24 h; 2: *E. coli* BL21(DE3) strain carrying pET-28a-TFP plasmid expressed for 24 h; 3: unpurified recombinant Dep_ZX1; 4: purified recombinant Dep_ZX1. (**c**) Spot tests of purified Dep_ZX1 on *K. pneumoniae* 111-2 lawn. Dep_ZX1 with 100, 10, 1, 0.1, 0.01, 0.001, and 0.0001 μg/mL (labeled on the left or right side of the spot) were measured for anti-capsule activity. (**d**) TEM picture of logarithmic *K. pneumoniae* 111-2. (**e**) TEM picture of logarithmic *K. pneumoniae* 111-2 treated with Dep_ZX1. (**f**) TEM picture of stationary growth *K. pneumoniae* 111-2. (**g**) TEM picture of stationary growth *K. pneumoniae* 111-2 treated with Dep_ZX1.

**Figure 6 pharmaceutics-14-01916-f006:**
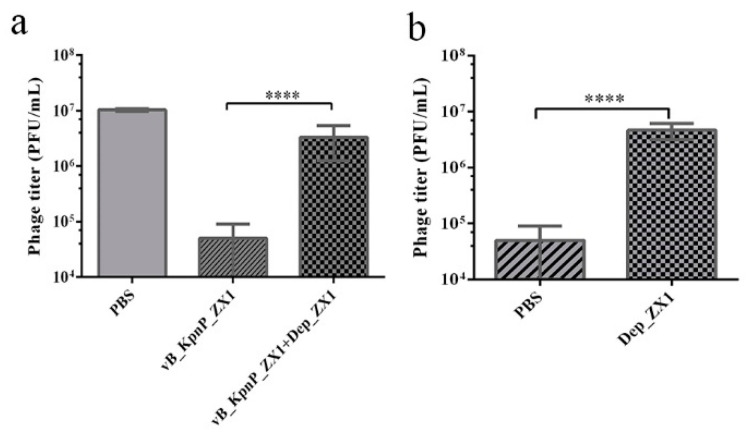
Depolymerase Dep_ZX1 reduces the adsorption of phage vB_KpnP_ZX1 to *K. pneumoniae* 111-2. (**a**) Phage titer in supernatant of competitive adsorption test between vB_KpnP_ZX1 and Dep_ZX1. (**b**). Phage titer in the supernatant of phage adsorption test after *K. pneumoniae* 111-2 treated with Dep_ZX1. *p* < 0.05 (*), 0.01 < *p* < 0.05 (**), 0.001 < *p* < 0.01 (***) and *p* < 0.001 (****).

**Figure 7 pharmaceutics-14-01916-f007:**
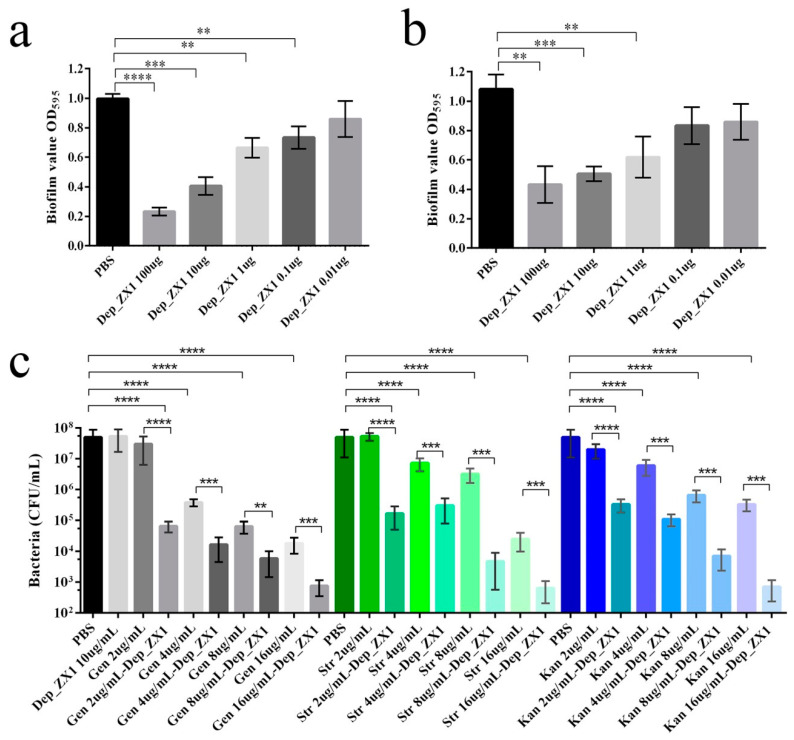
Antibiofilm and antibacterial activity of the depolymerase Dep_ZX1. (**a**) Dep_ZX1 and *K. pneumoniae* 111-2 was incubated in 96-well plates for 48 h. The biofilm was assessed by crystal violet staining, and the absorbance was measured at 595 nm. (**b**) Biofilm formed by *K. pneumoniae* 111-2 growth in 96-well plates for 48 h was treated with Dep_ZX1 for 3 h. (**c**) Biofilm formed by *K. pneumoniae* 111-2 growth in 96-well plates for 48 h was treated with Dep_ZX1 or antibiotics for 2 h. The viable bacterial count was determined on LB agar plates. *p* < 0.05 (*), 0.01 < *p* < 0.05 (**), 0.001 < *p* < 0.01 (***) and *p* < 0.001(****).

**Figure 8 pharmaceutics-14-01916-f008:**
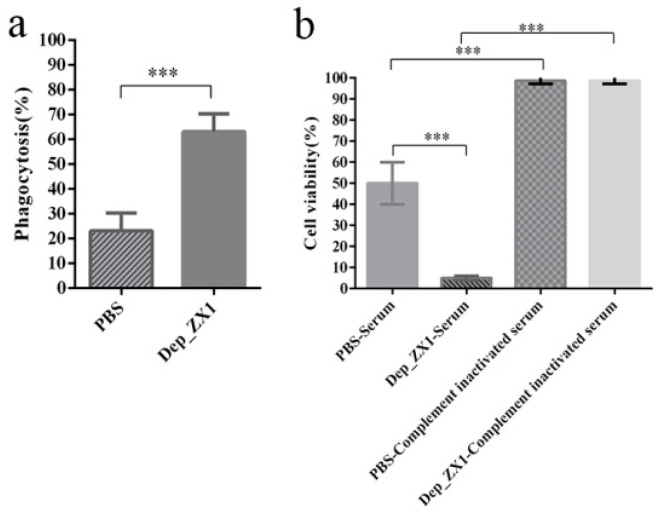
Depolymerase Dep_ZX1 improves the phagocytosis and serum sensitivity of *K. pneumoniae* 111-2. (**a**) Phagocytosis of phagocytes to *K. pneumoniae* 111-2 treated with PBS or Dep_ZX1. (**b**). The cell survival rate of *K. pneumoniae* 111-2 treated with PBS or Dep_ZX1 incubated with serum or complement-inactivated serum. *p* < 0.05 (*), 0.01 < *p* < 0.05 (**), 0.001 < *p* < 0.01 (***) and *p* < 0.001 (****).

**Figure 9 pharmaceutics-14-01916-f009:**
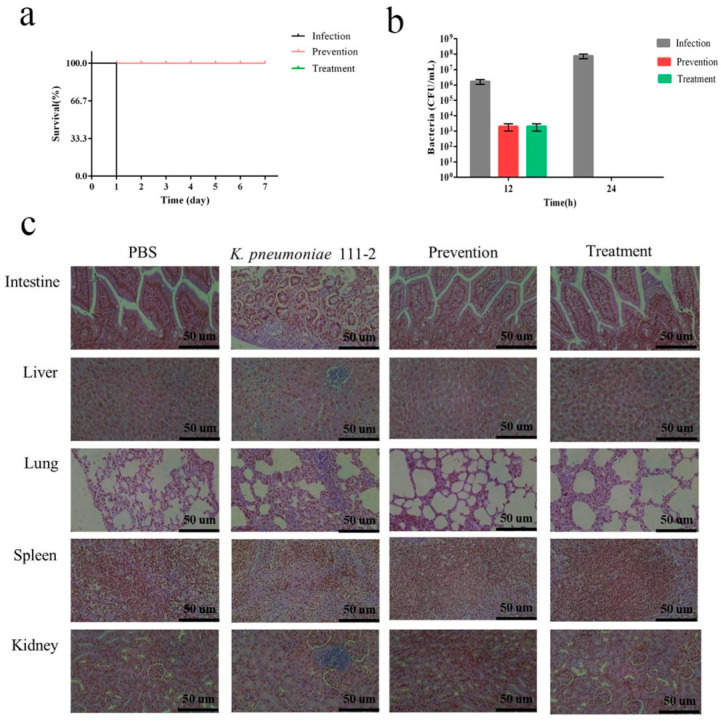
Protective and rescue effects of depolymerases Dep_ZX1 (50 µg/mice) in mice challenged with *K. pneumoniae* 111-2. (**a**) Survival rate of mice. (**b**) Number of bacteria in the blood of mice. (**c**) Pathological slices of small intestines, livers, lungs, spleens and kidneys in mice, protective or rescued by Dep_ZX1 after 48 h.

**Table 1 pharmaceutics-14-01916-t001:** Host range of phage vB_KpnP_ZX1.

*Klebsiella pneumoniae*	K Type	Lytic
O4	K1	−
AD1	K1	−
602-2	K3	−
A1	K20	−
N3	K54	−
AE3	K54	−
111-2	K57	+

+: Phage can form plaque; −: Phage cannot form plaque.

**Table 2 pharmaceutics-14-01916-t002:** The temperature and pH stability of depolymerase Dep_ZX1.

Minimum Effective Concentration/Treatment Conditions	40 °C	50 °C	60 °C	pH = 3	pH = 4	pH = 5	pH = 6	pH = 7	pH = 8	pH = 9	pH = 10	pH = 11
100 µg/mL	+	+	+	+	+	+	+	+	+	+	+	+
10 µg/mL	+	+	-	+	+	+	+	+	+	+	+	+
1 µg/mL	+	+	−	−	+	+	+	+	+	+	+	−
0.1 µg/mL	+	+	−	−	+	+	+	+	+	+	+	−

+: Dep_ZX1 can form degradation halo; −: Dep_ZX1 cannot form degradation halo.

## Data Availability

The genome sequence of phage vB_KpnP_ZX1 is available from the GenBank database (accession number MW722080).

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
