# Peer review of "Characterization of Novel Bacteriophage vB_KpnP_ZX1 and Its Depolymerases with Therapeutic Potential for K57 Klebsiella pneumoniae Infection"

_pharmaceutics, 2022, doi:10.3390/pharmaceutics14091916_

Round 1

Reviewer 1 Report

Dear Authors,

The manuscript ID: pharmaceutics-1889802_v1 entitled Characterization of Novel Bacteriophage vB_KpnP_ZX1 and its Depolymerases with Therapeutic Potential for K57 Klebsiella pneumoniae Infection” written by Ping Li, Wenjie Ma, Jiayin Shen and Xin Zhou is very creative and original.

Klebsiella pneumoniae has recently become a notorious virulent factor, due to the increase in the number of seriously infected patients. There is no doubt that its ability to evade the immune system, its increasing antimicrobial resistance and the emergence of hypervirulent pathotypes have become a major challenge in the medical field. Therefore, it is necessary to search for new antimicrobial solutions and therapies, and the use of bacteriophages as therapeutic agents becomes a very good idea.

The whole manuscript (Introduction, Materials and Methods, Results, Discussion and Conclusions) is properly organized. Introduction contains general data on the virulence of K. pneumoniae and phage therapy. Appropriate materials and methods were used to perform these studies. The obtained results are extensively documented, summarized in the form of figures, tables, schemes and photos, and properly interpreted. Based on the results, discussion and conclusions were drawn.

I agree with the Authors that the combination of novel bacteriophage, its depolymerases and antibiotics is expected to become a candidate drug in clinic for controlling K. pneumoniae infection or catheter-related infection caused by biofilm. This work provides many important clues to fight K. pneumoniae infection.

I have some suggestions in order to improve paper, which are the following:

1) Lines 282 – 284: „Antibiotic susceptibility test showed that K. pneumoniae 111-2 was resistant to ampicillin, sulfamethoxazole, tobramycin, azithromycin, erythromycin and vancomycin” – K. pneumoniae strains are naturally resistant to vancomycin so their susceptibility is not assessed in vitro.

2) K. pneumonic – K. pneumoniae

 In my opinion, the obtained results are valuable and manuscript is worth publishing in „Pharmaceutics.

With highest regards,

Author Response

Response to Reviewer 1 Comments

Point 1: Lines 282 – 284: “Antibiotic susceptibility test showed that K. pneumoniae 111-2 was resistant to ampicillin, sulfamethoxazole, tobramycin, azithromycin, erythromycin and vancomycin” – K. pneumoniae strains are naturally resistant to vancomycin so their susceptibility is not assessed in vitro.

Response 1: It has been revised in line 282-283 of the manuscript.

Point 2: K. pneumonic K. pneumoniae.

Response 2: The typing error of ‘K. pneumoniae’ has been revised in line 599 and 605 of the manuscript.

Reviewer 2 Report

The paper by Li et al reported the isolation and characterization of Klebsiella pneumonia phage vB_KpnP_ZX1. The authors then focus their attention on the putative depolymerase gene Dep_ZX1 of phage vB_KpnP_ZX1, and present both in vitro and in vivo evidence to suggest that this depolymerase is highly effective against K57 type K. pneumonia capsule. Overall, the paper is well written and is easy to follow. However, all the figures are of low quality, and the font size for the figure labels is very hard to read.

Other minor comments:

1) Abstract: “Phage vB_KpnP_ZX1, encoding three lysogen genes, repressor, anti-repressor and integrate,” should be “integrase” not integrate

2) Page 6 last paragraph: “Phage vB_KpnP_ZX1 forms transparent round plaque with a halo around on the bacterial lawn of K. pneumonia 111-2 (Figure 1a)”.

What does “transparent round plaque” mean? Is it turbid or clear? The low-quality photo in Figure 1a makes it very hard to see the halo.

3) Figure 1b, no scale bar, please modify the figure to include the scale bar

4) Page 7, section 3.2, “The thermal and pH stability analysis show that vB_KpnP_ZX1 maintained a medium tolerance range. Phage vB_KpnP_ZX1 viability show a high stable after treated at 299 4-40 °C for 1 h.” only 37°C -80°C data was present on Figure 2b, the 4°C to 37°C data is missing.

5) Page 7, last paragraph, “For example, when phages were added at the ratio of MOI=10, the growth trend of bacteria was inhibited in the first 60 min, the number of bacteria sharply decline in 60-120 min and 306 increased again after 120 min.”

Figure 2d shows a classic example of infection curves expected from a temperate phage, maybe it is worth commenting on that point in the main test, see

https://www.mdpi.com/1424-8247/14/10/998

https://www.frontiersin.org/articles/10.3389/fmicb.2017.01386/full

6) Page 8, section 3.2, “No virulence gene and drug resistance gene was detected. Gp23 repressor, gp28 anti-repressor, gp77 integrate were annotated as lysogen related gene.” Again, should be “integrase” not integrate

7) Page 8, section 3.2, the following paragraph: “The whole-genome dot plotting analysis shows that sixteen phages from different genus were classified into 5 clusters, within which phage vB_KpnP_ZX1, SopranoGao, SJM3 and LASTA belong to Podoviridae family, Lastavirus genus (Figure 3c). In addition, phylogenetic analysis of the phage terminase large subunit confirms the same result (Figure 3d).” Figure 3c should be Figure 3d, and Figure 3d should be Figure 3c

8) Page 9, last paragraph, “The low similarity of TFP suggests differences in the adsorption receptor and host range between phage vB_KpnP_ZX1 and the others.”

What is known about the host range of the related phages mentioned above? It is worth discussing them if known.

9) Page 11, Figure 5c, why labels on both sides of the photo? What does that mean?

10) Figure 5d, e, f, g, no scale bar, again, please modify the figure to include the scale bar

Author Response

Response to Reviewer 2 Comments

Point 1: Abstract: “Phage vB_KpnP_ZX1, encoding three lysogen genes, repressor, anti-repressor and integrate,” should be “integrase” not integrate.

Response 1: The typing error of ‘integrase’ has been revised in line 17 of the manuscript.

Point 2: Page 6 last paragraph: “Phage vB_KpnP_ZX1 forms transparent round plaque with a halo around on the bacterial lawn of K. pneumonia 111-2 (Figure 1a)”.

What does “transparent round plaque” mean? Is it turbid or clear? The low-quality photo in Figure 1a makes it very hard to see the halo

Response 2: A clearer photo of plaque was used in Figure 1a. “Phage vB_KpnP_ZX1 forms turbid round plaque with a halo around on the bacterial lawn of K. pneumoniae 111-2 (Figure 1a)’. It has been revised in line 284 of the manuscript.

Point 3: Figure 1b, no scale bar, please modify the figure to include the scale bar

Response 3: A clearer scale bar is added in figure 1b.

Point 4: Page 7, section 3.2, “The thermal and pH stability analysis show that vB_KpnP_ZX1 maintained a medium tolerance range. Phage vB_KpnP_ZX1 viability show a high stable after treated at 299 4-40 °C for 1 h.” only 37°C -80°C data was present on Figure 2b, the 4°C to 37°C data is missing.

Response 4: The viability data of phage vB_KpnP_ZX1 at 4, 10, 20 and 30°C are added in figure 2b.

Point 5: Page 7, last paragraph, “For example, when phages were added at the ratio of MOI=10, the growth trend of bacteria was inhibited in the first 60 min, the number of bacteria sharply decline in 60-120 min and 306 increased again after 120 min.”

Figure 2d shows a classic example of infection curves expected from a temperate phage, maybe it is worth commenting on that point in the main test, see

https://www.mdpi.com/1424-8247/14/10/998

https://www.frontiersin.org/articles/10.3389/fmicb.2017.01386/full.

Response 5: Although phage vB_KpnP_ZX1 encodes an integrase, we have not isolated lysogenic K. pneumoniae 111-2 during the propagation of phage. Furthermore, no lysogenic K. pneumoniae 111-2 was isolated after overexpression of the integrase encoded by phage vB_KpnP_ZX1 during its propagation (data not published). Therefore, we believe that phage vB_KpnP_ZX1 cannot be integrated into the genome of K. pneumoniae 111-2. We speculate that the increase in the number of bacteria at the later stage of phage vB_KpnP_ZX1 infection is caused by the production of phage resistant strains rather than the lysogenesis of bacteria. It has been revised in line 307-311 of the manuscript.

Point 6: Page 8, section 3.2, “No virulence gene and drug resistance gene was detected. Gp23 repressor, gp28 anti-repressor, gp77 integrate were annotated as lysogen related gene.” Again, should be “integrase” not integrate

Response 6: The typing error of ‘integrase’ has been revised in line 319 of the manuscript.

Point 7: Page 8, section 3.2, the following paragraph: “The whole-genome dot plotting analysis shows that sixteen phages from different genus were classified into 5 clusters, within which phage vB_KpnP_ZX1, SopranoGao, SJM3 and LASTA belong to Podoviridae family, Lastavirus genus (Figure 3c). In addition, phylogenetic analysis of the phage terminase large subunit confirms the same result (Figure 3d).” Figure 3c should be Figure 3d, and Figure 3d should be Figure 3c

Response 7: The order of figure 3c and figure 3d has been changed.

Point 8: Page 9, last paragraph, “The low similarity of TFP suggests differences in the adsorption receptor and host range between phage vB_KpnP_ZX1 and the others.”

What is known about the host range of the related phages mentioned above? It is worth discussing them if known.

Response 8: Phage vB_KpnP_ZX1 can specifically infect K57 type K. pneumoniae in this study. Unfortunately, the host range of phage SopranoGao, SJM3 and LASTA has not been reported so far. It has been revised in line 349-351 of the manuscript.

Point 9: Page 11, Figure 5c, why labels on both sides of the photo? What does that mean?

Response 9: Figure 5c: Spot tests of purified Dep_ZX1 on K. pneumoniae 111-2 lawn. Dep_ZX1 with 100, 10, 1, 0.1, 0.01, 0.001 and 0.0001 μg/mL (labeled on the left or right side of the spot) were measured for anticapsule activity. It has been revised in line 425-427 of the manuscript.

Point 10: Figure 5d, e, f, g, no scale bar, again, please modify the figure to include the scale bar

Response 10: A clearer scale bar is added in figure 5d, e, f, g.
